# Biological Treatment of Real Textile Effluent Using *Aspergillus flavus* and *Fusarium oxysporium* and Their Consortium along with the Evaluation of Their Phytotoxicity

**DOI:** 10.3390/jof7030193

**Published:** 2021-03-09

**Authors:** Mohamed T. Selim, Salem S. Salem, Asem A. Mohamed, Mamdouh S. El-Gamal, Mohamed F. Awad, Amr Fouda

**Affiliations:** 1Department of Botany and Microbiology, Faculty of Science, Al-Azhar University, Nasr City, Cairo 11884, Egypt; mohamed.talaat@azhar.edu.eg (M.T.S.); salemsalahsalem@azhar.edu.eg (S.S.S.); mamdouh.elgamal@azhar.edu.eg (M.S.E.-G.); 2National Research Center, El-Behouth St. (Former El-Tahrir Str.), Dokki, Giza 12622, Egypt; aalfarouk@yahoo.com; 3Department of Biology, College of Science, Taif University, P.O. Box 11099, Taif 21944, Saudi Arabia; m.fadl@tu.edu.sa

**Keywords:** decolorization, textile wastewater, fungi, water scarcity, GC–MS, plants irrigation

## Abstract

Twenty-one fungal strains were isolated from dye-contaminated soil; out of them, two fungal strains A2 and G2-1 showed the highest decolorization capacity for real textile effluent and were, hence, identified as *Aspergillus flavus* and *Fusarium oxysporium* based on morphological and molecular methods. The highest decolorization percentage of 78.12 ± 2.1% was attained in the biotreatment with fungal consortium followed by *A. flavus* and *F. oxysporium* separately with removal percentages of 54.68 ± 1.2% and 52.41 ± 1.0%, respectively. Additionally, ultraviolet-visible spectroscopy of the treated effluent showed that a maximum peak (*λ*_max_) of 415 nm was reduced as compared with the control. The indicators of wastewater treatment efficacy, namely total dissolved solids, total suspended solids, conductivity, biological oxygen demand, and chemical oxygen demand with removal percentages of 78.2, 78.4, 58.2, 78.1, and 77.6%, respectively, demonstrated a considerable decrease in values due to fungal consortium treatment. The reduction in peak and mass area along with the appearance of new peaks in GC-MS confirms a successful biodegradation process. The toxicity of treated textile effluents on the seed germination of *Vicia faba* was decreased as compared with the control. The shoot length after irrigation with effluents treated by the fungal consortium was 15.12 ± 1.01 cm as compared with that treated by tap-water, which was 17.8 ± 0.7 cm. Finally, we recommended the decrease of excessive uses of synthetic dyes and utilized biological approaches for the treatment of real textile effluents to reuse in irrigation of uneaten plants especially with water scarcity worldwide.

## 1. Introduction

Industrialization processes are considered as one of the main aspects for enhancing economic growth worldwide; however, the effluents released from these processes are major contributors to pollution and eco-toxicity [1,2]. Colored effluents are produced because all industrial processes, such as textile, dye manufacturers, pharmaceuticals, foods, plastics, cosmetics, leathers, rubbers, papers, and pulp, contain dyes [3,4,5]. The textile industry involves a broad range of machinery, raw substances, and different processes to manufacture end products with specific shapes and high qualities. This industry is considered the oldest and popular industry worldwide, starting from 3000 BC, and provides the largest number of employments [6]. The effluents released from the textile industry are considered as one of the highest liquid pollutants, based on the published literature. About 280,000 tons of textile dyes are discharged as wastes in textile effluents every year worldwide [7,8]. A massive quantity of water is consumed during the textile processing steps such as washing, dyeing, seizing, and others. Approximately, 10–20 L of pure water is consumed for dyeing 1 kg of fabric [9]. Annually, the environment and different waterways receive about 1000 tons of dyes besides other pollutants such as aromatic amines, flame retardants, heavy metals, bisphenols, and so on without any treatment [10]. As a result, these effluents destroy the soil and aquatic ecosystems by altering the microbial community and reducing the quality of surface and groundwater [11]. The problem related to textile effluents is not only attributed to the presence of dyes but also to the presence of highly stable and non-degradable pollutants [7]. Recently, textile wastewater treatment has captured more attention because of the strict legislation concerning the discharge of this effluent in the ecosystem [7]. Approximately, one million tons of dyes are produced every year worldwide, out of which nearly 50% are discharged in the effluent streams or end up in landfills [12]. The environmental problems correlated with textile effluents occur because of the use of various non-degradable dyes such as azo dyes, high biochemical and chemical oxygen demand (BOD and COD), high pH, salts, and increased levels of suspended solids [13,14,15]. Moreover, textile wastewater contains other pollutants, such as color residues, inorganic compounds, catalytic chemicals, dye waste, and cleaning solvents [16,17]. The negative impacts of the discharge of dyes into environment without any treatment can be related to adverse effects on the photosynthetic activity, which restricts the access of light and causes a shortage of oxygen, further decreasing the survival rates of flora and fauna [18,19]. Additionally, different dyes discharged from industries are highly toxic and considered a mutagenic agent. Therefore, the major challenge is to establish new and viable approaches to degrade these dyes because they are difficult to degrade by traditional methods. Importantly, these dyes exert carcinogenic and toxigenic effects on human and aquatic systems [19,20].

Several conventional methods have been used for the treatment of textile effluents such as physical, chemical, and some modified techniques, including electrolysis, adsorption, photo-ionization, ozonation, membrane filtrations, and oxidation [21,22,23]. However, these methods have some restrictions, such as lesser capability, higher costs, and resulting hazardous by-products [24]. Nowadays, effective biological treatment processes are of great value due to their eco-friendly, low cost, and minor sludge-giving properties [15,25,26]. Different metabolites produced by microbes have a high potentiality in various biological activities [15]. The potentiality of fungi in biodegradation is known as mycoremediation. Fungi are well known for their superior abilities to produce a well-built variety of extracellular proteins and other organic compounds [27,28]. Fungi easily adapt to severe environmental constraints and can be easily manipulated with different problems [15]. Moreover, the requirement of nitrogen-limiting conditions and a long growth cycle leads to prolonged hydraulic retention times. Importantly, preserving the dominance of fungal cultures in the reactor system is a major challenge that prevents the use of fungi for full-scale applications [7,29].

The objective of this study is to explore and compare the decolorization efficiencies of crude textile effluents by two native fungal species, namely *Aspergillus flavus* and *Fusarium oxysporium*, isolated from the soil contaminated with textile dyeing. Additionally, this study also assesses the effects of different parameters such as pH, inoculum sizes, fungal consortium, and different sources of nitrogen and carbon on the decolorization of dyes. Moreover, this study’s other main objectives include the comparison of biodegradable end products with the control as well as the investigation of the reusability of treated effluents in plant irrigation.

## 2. Materials and Methods

### 2.1. Materials

The contaminated soil samples (GPS: 30°58′10.8156″ N, 31°10′5.0988″ E) used for fungal isolation were collected from the textile industry company located in El Mahalla El Kubra, Gharbia Governorate, Egypt. The final effluents from the same textile industry were obtained as a model to study the efficacy of fungal isolates in crude textile wastewater treatment. Plastic bottles were used for storing the textile wastewater samples instead of glass bottles because wastewater containing hazardous substances react with sodium in the glass matrix [30].

### 2.2. Fungal Isolations

The collected contaminated soil samples were screened for the isolation of the fungal species on sabouraud dextrose agar (SDA) media. One gram of each soil sample was transferred to 9 mL of distilled water and serially diluted up to 10^−7^. About 0.1 mL of the sixth dilution was inoculated onto the SDA plates supplemented with 100 mg·L^−1^ of chloramphenicol to suppress the bacterial growth. The plates were incubated at 28 ± 2 °C for 5–7 days. The fungal growth was checked for purity, and the purified isolate was inoculated in the culture slant for preservation at 4 °C for further study [31]. In total, 21 purified fungal species were isolated and subjected to primary identification based on morphological and cultural characteristics.

### 2.3. Screening for the Biodegradation Efficacy of Crude Textile Wastewater

The optical density of the crude textile effluent was observed at different wavelengths ranging between 200 and 800 nm using spectrophotometer UV (Shimadzu 1800). The wavelength at which the textile effluent showed the maximum absorbance (*λ*_max_: at 415.0 nm) was used as a standard for all treatments. In total, 21 purified fungal isolates were screened for their wastewater-degrading activity by inoculating in 100 mL of crude textile wastewater without any other supplementation. The treatment was conducted in sterilized and non-sterilized wastewater as follows: one fungal disc (7 mm diameter) was cut from the periphery of the fifth old culture and inoculated separately in a conical flask containing 100 mL of sterile and non-sterile wastewater. All the flasks were incubated at room temperature on a shaker at 150 rpm. The control (sterilized and non-sterilized wastewater without fungal inoculation) treatment was run with the above experiment under the same condition. The experiment was performed in triplicates. About 2 mL of wastewater from each treatment was withdrawn at the end of each incubation period and centrifuged at 10,000 rpm for 10 min. The clear supernatant was used for measuring the absorbance at *λ*_max_ of 415.0 nm after 4, 7, 9, and 11 days, and the decolorization percentages were calculated according to the following equation [15]:$$D\;(\%)=\frac{\mathrm{Dye}\;(i)-\mathrm{Dye}\;(I)}{\mathrm{Dye}\;(i)}\times 100$$
where D is the decolorization percentage (%); Dye (i) is the initial absorbance; Dye (I) is the final absorbance.

### 2.4. Molecular Identification of the Most Potent Fungal Isolates

The two most potent fungal isolates were genetically identified through genomic DNA and by amplifying the internal transcribed spacer (ITS) region. For the ITS-based sequencing, the genomic DNA was extracted from 100 mg of mycelium. DNA extraction was conducted by the protocol of Gene Jet Plant genomic DNA purification Kit (Thermo) #k0791. The primers used were ITS1 (5′-CTTGGTCATTTAGAGGAAGTAA-3′) and ITS4 (5′-TCCTCCGCTTATT GATATGC-3′) [32,33]. The amplification (PCR) protocol used was Maxima Hot Start PCR Master Mix (Thermo) #k0221 by Sigma Scientific Services Company (Cairo, Egypt) as follows: the PCR mixture (50 μL) contained Maxima Hot Start PCR Master Mix (Thermo), 0.5 μM of each primer, and 1 μL of extracted fungal genomic DNA. The PCR was performed in a DNA Engine Thermal Cycler with hot starting performed at 94 °C for 3 min, followed by 30 cycles of 94 °C for 0.5 min, 55 °C for 0.5 min, and 72 °C for 1 min, followed by a final extension performed at 72 °C for 10 min. The samples were sequenced by ABI 3730 × 1 DNA sequencer by forward and reverse primers by GATC Company (Germany). The ITS sequences of the identified fungal isolates were compared against the GenBank database using Clustal × 1.8 software package (http://www.clustal.org/clustal2, accessed on 17 November 2010). Phylogenetic analysis was conducted by the neighbor-joining method using MEGA v6.1 software, with confidence testing by bootstrap analysis (1000 repeats) [34].

### 2.5. Optimization Conditions of Biodecolorization Process Using the Most Potent Fungal Isolate

#### 2.5.1. Effects of Different Incubation Periods, pH Values, and Inoculum Size on the Decolorization Percentages (%)

To investigate the effects of different pH values on the decolorization process, fungal strains A2 and G2-1 were inoculated in wastewater adjusted at different pH values of 6.0, 7.0, 8.0, 9.0, 10.0, and 11.0. The decolorization percentages were detected at different incubation times (3.0, 4.0, 5.0, 6.0, 7.0, 8.0, and 9.0 days). The decolorization percentages were estimated under different inocula sizes represented by 1.0, 2.0, 3.0, and 4.0 discs. Each disc with a heavy spore fungal growth diameter of 7 mm inoculated in 100 mL of crude textile wastewater. At the end of each incubation period, 2 mL of each treatment were withdrawn and centrifuged at 10,000 rpm for 10 min and decolorizations (%) were assessed using the above equation.

#### 2.5.2. Effects of Different Carbon and Nitrogen Sources on the Dye Decolorization by *Aspergillus flavus* (Strains A2) and *Fusarium oxysporium* (Strain G2-1)

To evaluate the effects of different carbon and nitrogen sources on dye decolorization, different carbon sources were added into wastewater containing 2% concentration with equimolecular level for each sugar. The media without carbon source was used as a control. The carbon sources were represented by glucose, sucrose, starch, bagasse, and cellulose. Similarly, with the equivalent amounts of nitrogen sources located at 0.5%, the effects of different organic and inorganic nitrogen sources such as sodium nitrate, sodium nitrite, ammonium sulfate, urea, peptone, and yeast extract on the decolorization of crude wastewater were evaluated. In each case, all the previously mentioned optimal conditions of pH, incubation period, and inoculation size were considered. At the end of each incubation period, 2 mL of each treatment were withdrawn and centrifuged at 10,000 rpm for 10 min and decolorizations (%) were assessed using the above equation.

### 2.6. Analysis of Crude Textile Wastewater before and after Treatment

Physicochemical characteristics of the crude textile wastewater analysis before and after the treatment processes were estimated for pH, conductivity, color removal, total dissolved solids (TDS), total suspended solids (TSS), chemical oxygen demand (COD), and biological oxygen demand (BOD). These analyses were conducted according to the standard methods recommended by the American Public Health Association [35], which are recommended by the Egyptian Law 93/1962 Dec. 44/2000 to measure the reduction in these parameters.

#### 2.6.1. GC–MS Spectroscopy for Untreated/Treated Wastewater

The untreated/treated textile wastewater was analyzed using gas chromatography and mass spectroscopy (GC–MS) (The Agilent 5975C Series GC/MSD and The Agilent 7890A Gas Chromatograph). GC–MS is an effective combination for chemical analysis. The GC analysis separates compounds into complex mixtures, and the MS analysis determines the molecular weight and ionic fragments of individual components, further aiding in the identification of those compounds. Helium was used as the carrier gas at a flow rate of 1 mL min^−1^. The injector temperature was maintained at 280 °C, whereas the oven conditions were maintained at 80 °C for 2 min, followed by an increase to 200 °C based on a rate of 10 °C at 7 min, followed by a further increase to 280 °C based on a rate of 20 °C/min. The compounds were identified based on their mass spectra and using the National Institute of Standards and Technology (NIST) library [36,37]. The numerous standards of the American Society for Testing and Materials that cover GC/MS are also utilized for routine determinations.

#### 2.6.2. Phytotoxicity Study

The treated and untreated textile effluents were evaluated for their toxic effect on the germination of *Vicia faba* L. It is one of the important vegetable crops. The seeds of *Vicia faba* L. were obtained from the Agricultural Research Center, Cairo, Egypt. Seeds having a uniform size were chosen for the experimental study. The seeds were surface-sterilized by soaking in 2.5% sodium hypochlorite for 3 min and then washed with sterile distilled water 5 times [38]. Sterilized seeds were soaked in distilled water for 24 h before starting the experiment. The experiments were conducted in triplicates with each Petri plate containing Whatman filter paper loaded with four uniformly seedling seeds. The experiment was irrigated as needed with treated and untreated effluents. A control was set up with distilled water. These were incubated at room temperature for seven days in dark conditions. The length of plumule (shoot) and radicle (root) was recorded for comparison with the control.

### 2.7. Statistical Analysis

All results presented in this study are the means of three independent replicates. Data were subjected to analysis of variance (ANOVA) by a statistical package SPSS v17. The mean difference comparison between the treatments was analyzed by the Tukey HSD test at *p* < 0.05.

## 3. Results and Discussion

### 3.1. Fungal Isolation and Identification

In this study, twenty-one fungal isolates were isolated from contaminated soil samples and subjected to primary identifications using standard keys. Data showed that the fungal isolates belong to *Aspergillus* spp. (coded as A1, G1-2, D1-1, F2, and G2-3), *Aspergillus flavus* (A2), *Aspergillus niger* (coded as D1-3), *Penicillium* spp. (coded as B1-1, D1-2, E1, E2, F1, and G1-1), *Rhizopus* spp. (coded as B2-1, F3, and G1-3), *Alternaria* spp. (coded as C1 and G2-2), *Fusarium* sp. (E3), *Fusarium oxysporium* (G2-1), and *Mucor* sp. (G1-2). Data analysis showed different capabilities of fungal species to utilize crude textile wastewaters without any supplementation at different conditions (sterile and non-sterile). In sterile conditions (Figure 1A), the highest decolorization rates of textile wastewater were 25.73 ± 0.02% and 23.13 ± 0.64% for the fungal isolates A2 and G2-1, respectively. In contrast, in non-sterile conditions (Figure 1B), the highest decolorization rates were 26.06 ± 1.35% and 23.45 ± 0.36% for the same fungal isolates. Furthermore, the obtained data revealed that the maximum decolorization of textile effluent by the two fungal isolates was achieved at 7 days of incubation period and the decolorization (%) decreased gradually after 9 days. From an industrial overview, the results in the case of sterile wastewater were slightly similar to those obtained by non-sterile wastewater. Therefore, a less costly non-sterile condition was selected for further investigation in this study.

The genetic identification of the two most promising fungal isolates, namely A2 and G2-1, was accomplished by the sequencing of ITS genes. The molecular identification based on the ITS sequencing analysis for isolates A2 and G2-1 were like *A. flavus* (accession number: MN095180) and *F. oxysporum* (accession number: MG274310) with similarity percentages of 98.5% and 99.5%, respectively (Figure 2). The sequences obtained in this study were deposited in GeneBank under accession numbers MW133778 and MW133779 for *A. flavus* and *F. oxysporum*, respectively.

### 3.2. Effect of Different Incubation Periods and pH Values on Dye Decolorization Percentages

The maximum decolorization of textile effluent by *A. flavus* (A2) and *F. oxysporum* (G2-1) was recorded at pH 8.0 and 6.0, respectively. The results showed that the maximum decolorization percentage of wastewater by *A. flavus* has reached 26.2 ± 1.3% at pH 8.0 after 7 days, while the maximum decolorization percentage of wastewater by *F. oxysporum* has reached 27.8 ± 0.02% at pH 6.0 after 7 days (Table 1). In addition, data showed that significant decolorization of textile effluent by fungal strains was achieved at 7 days of the incubation period. After 7 days, the decolorization decreased gradually, which may be attributed to the efflux mechanism of dye from the fungal cells to reduce dye toxicity.

pH values have a direct impact on enzymatic activities and physiological performance of microbial cells, and hence control the transport of various nutrients in/out of the cell membrane. Therefore, the pH of the decolorization medium has a marked effect on the process [39]. In this study, the maximum decolorization of textile effluents was recorded at pH values 6 and 8; any increase or decrease in pH values reduces the decolorization efficiency. Data of this study supports that the decolorization of the textile effluents reaches the maximum value in weak alkaline or weak acidic medium. It is achievable that pH change affects the transport of dye particles crossway the cell membrane of microbes, which is reflected as the rate-limiting stage for decolorization [40]. The results in the present study were constant with those obtained by Asha et al. [41] and Srinivasan [42] reported that the optimum dye decolorization of textile effluent by *Enterobacter asburiae* and azo dye Drimaren Red by *Aeromonas hydrophila* and *Lysinibacillus sphaericus,* respectively, were achieved at pH 8. Moreover, Ketut Sudiana et al. [43] reported that the optimum pH value for decolorization of Remazol Black B by *Ganoderma* sp. was 6. Additionally, the maximum decolorization of Synozol red HF-6BN by *A. niger* was done at a pH value of 6 [44]. Although most azo-reductase enzymes have an optimum pH value of 7, some alkali-thermostable azo-reductase showed maximum decolorization and optimum pH within the range of 8–9 [45]. The decolorization of dyes at higher concentrations using fungi was preferred in an acidic condition because they facilitate enzymatic activities and increase adsorption by the fungal cell wall [46]. In contrast, Dave et al. [47] reported that the fluctuation in the pH values from neutral to slightly alkaline has a very little effect on the dye decolorization as compared to acidic conditions. Moreover, Nor et al. [48] reported that acidic pH values are generally enriched and maintain well the culture of microorganisms for optimal decolorization. Mostafa et al. [19] reported that the surface electrical charge of fungal biomass and the ionic forms of dye pollutants can be controlled by the pH value of the solution. For instance, the optimum removal of acidic dye can be achieved at acidic conditions because there was an increase in the protonation of weak base group (biomass) that bounded and degraded the anionic group of acidic dye. Finally, the optimum pH for the decolorization of dyes highly depends on the type of fungi, dyes, medium, and environmental conditions [19].

The maximum decolorization of dyes was accomplished at 7 days of the incubation period. After 7 days of incubation, the decolorization decreased gradually. This decrease might occur due to the efflux mechanism of dye from the fungal cells to reduce dye toxicity. This study results were similar to those obtained by Hefnawy et al. [49] who found that the maximum decolorization of direct blue dye by *A. flavus* was obtained at 7 days of incubation and the decolorization decreased gradually after 7 days of incubation. Additionally, this phenomenon could be attributed to the enzymatic biodegradation activity along with the physical binding of dye on fungal biomass [49]. Interestingly, the obtained results are compatible with that of Salem et al. [15], who reported that the maximum decolorization of reactive yellow and reactive red by *A. niger* was obtained at 7 days and the decolorization percentages were decreased gradually after that. Salem et al. [15] clarified this phenomenon by the accumulation of dye products that might have hindered growth and the metabolizing potential of fungi.

### 3.3. Effect of Different Inocula Sizes (Discs) on Decolorization Rate

The results represented graphically in Figure 3A show that the decolorization percentages were increased by increasing the inocula size for each isolate. The highest decolorization of textile effluent was recorded at the inocula size of 3 discs/100 mL crude textile effluent. The decolorization percentages due to treatment by three discs of *A. flavus* and *F. oxysporium* separately were 32.9% and 33.9%, respectively. The decolorization percentages decreased by increasing the inocula size up to 3 discs.

The obtained results were compatible with the results obtained by Salem et al. [15], who reported that the decolorization percentages of reactive yellow (4GL) and reactive red (4BL) dyes were 57.6% and 39.9%, respectively, due to treatment with two discs (7 mm in diameter) of *A. niger*. Boda et al. [50] reported that the decreasing value of COD and increasing the color removal of textile wastewater depend on the selected microbes and the concentration of inocula size. Hameed and Ismail [51] reported that the decolorization percentages of reactive blue azo dye were increased due to inoculation with 10% (*v*/*v*) of bacterial consortium isolated from activated sludge.

### 3.4. Effect of Different Carbon and Nitrogen Sources on the Dye Decolorization

The highest decolorization and biodegradation of azo dyes were usually achieved through co-metabolism in the presence of additional nutritional support such as glucose and yeast extract [52]. Several economically cheap and raw carbon sources, such as molasses, bagasse, and jaggers have been used to improve decolorization and reduce the cost of the process [47]. Hence, in this study, different co-substrates such as sucrose, starch, glucose, cellulose, and baggas (g·L^−1^) were supplemented in textile effluent as additional carbon sources. The best carbon source was glucose, which caused the highest decolorization of dyes followed by sucrose and starch. The highest decolorization percentages of textile effluents were 40.2% and 39.1% in the presence of glucose and inoculation by *A. flavus* (A2) and *F. oxysporum* (G2-1), respectively (Figure 3B). This finding was reasonable because glucose is more effortlessly metabolized than other sugars and therefore can be utilized readily as a source of energy. Additionally, it was hypothesized that the rate of formation of reduced nucleotides, such as NADH and FADH, was increased when glucose was supplemented as the carbon source. The presence of these redox mediators, which are involved in the reduction of azo bond, is essential to promote a better decolorization efficiency [52].

The lowest decolorization percentages were recorded for textile effluents in the presence of cellulose (34.9 ± 1.5% for *A. flavus* and 34.7 ± 0.5% for *F. oxysporum*). In the presence of bagasse, the decolorization percentages were 34.9 ± 0.0% and 34.9 ± 0.0% for *A. flavus* and *F. oxysporum*, respectively. This study results were similar to those obtained by Mostafa et al. [19], who showed that glucose and fructose were the best carbon sources to remove dyes by *Cylindrocephalum aurelium* in textile wastewater.

The biodecolorization activity of textile effluents increased in the presence of yeast extract as additional nitrogen sources for two fungal strains after seven days. The highest decolorization percentages of textile effluent reached up to 54.5 ± 0.9% and 53.4 ± 0.9% in the presence of yeast extract by using *A. flavus* (A2) and *F. oxysporum* (G2-1), respectively. However, the lowest decolorization percentages of textile effluent reached up to 44.8 ± 0.1% and 43.8 ± 0.0% for *A. flavus* and *F. oxysporum*, respectively, in the presence of ammonium sulfate after 7 days (Figure 3C).

Imran et al. [53] reported that yeast extract has a remarkable capability of enhancing enzyme activity as compared to other commonly used sources. Fungi can utilize a variety of nitrogen sources, including inorganic nitrate, ammonium salts, and some organic compounds especially amino acids. The different percentages reflected the capacities of the filtrates in removing dyes with diverse chemical structures. This might be attributed to the differences in electron distribution, charge density, or steric factors [19]. Consistent with this study, the maximum decolorization percentages of Remazol Black B were achieved in the presence of glucose (87.5%) and yeast extract (83.3%) from treatment with *Pseudomonas aeruginosa* [54]. Carbon and nitrogen sources were used to enhance enzyme production because they contain essential minerals and nutrients to be used as energy for growth [19].

### 3.5. Assessment of the Decolorization Efficacy of Textile Wastewater by the Two Fungal Strains Separately or in Consortium under All the Optimized Factors

The biodegradation and decolorization of textile wastewater using the two fungal strains separately or in consortium under the optimized factors were investigated to maximize the fungal decolorization capacity. The highest decolorization percentage reached 78.1 ± 2.1% for crude textile wastewater after 7 days in consortium as compared with the individual fungal strain (54.7 ± 1.2% for *A. flavus* A2 and 52.4 ± 1.0% for *F. oxysporium* G2-1) (Figure 4A).

The consortium between organisms exhibits a high efficacy in azo dye degradation. For example, the consortium between *Aspergillus* sp. and *Pseudomonas* sp. showed high detoxification of Rubine dye [55]. The same consortium exhibits a high degrading efficacy (98%) of textile effluents comprising reactive dyes and disperse azo dyes [55]. Various bacterial isolates showed high azo dye degradation when applied together as a consortium rather than individually. Many reports are available on the dye-degrading assays using bacterial consortium. For the first time, Nigam et al. reported that the mixing of *Micrococcus luteus*, *Micrococcus* sp., and *Paenibacillus polymyxa* showed high azo dyes degradation efficacy, but did not show any activities in the individual state. Moreover, a consortium of four bacterial isolates, namely *Pseudomonas putida, Pseudomonas fluorescens, Bacillus cereus*, and *Stenotrophomonas acidaminiphila*, showed the degradation of Acid Red 88 within 24 h, but when inoculated individually, each isolate took more than 72 h for degradation [56]. In another study, the effects of isolates *Klebsiella* sp., *Bacillus* sp., and *Clostridium* sp. showed a good degradation ability under aerobic conditions, whereas the same consortium showed no change under anaerobic conditions [57].

### 3.6. UV-Visible Spectrum for Textile Wastewater before and after Optimization

The full UV–visible spectrum of the decolorized samples could reveal the mode of decolorization. A decrease in the visible region (200–800 nm) indicates a decrease in the dye color, whereas the complete disappearance of the peaks indicates that the removal of color occurs through the breakdown of the azo bond, which is responsible for dye color. A UV–Vis spectrophotometer was used to monitor the change in absorbance as previously described in another research works [20,58]. In this study, a decrease in the peak area at ~415 nm in the corresponding UV–Vis spectrum of the treated effluent strongly indicates the degradation of dye effluent due to fungal action (Figure 4B).

According to UV–Vis spectroscopic analysis of the textile effluent before and after treatment with *F. oxysporium*, *A. flavus*, and their consortium, we noted that the absorbance peaks in the visible region decreased without any shift in *λ*_max_, thereby indicating the potentiality of fungal strain at individual or consortium state in the decolorization of the solution. This study results were compatible with those obtained by Chaieb et al. [59], who reported that the absorbance peaks of azo dyes (Cong red, Evans blue, and Eriochrome Black T) in the visible region were decreased without any shift in *λ*_max_ because of treatment with *Staphylococcus lentus*. The decreased or absence of the peaks in the visible region is probably due to the color removal and formation of aromatic amines resulting from the cleavage of the azo bond [60]. Nouren et al. [61] reported that the untreated direct yellow dye showed a wide peak at 402 nm, whereas treatment with direct yellow dye by citrus lemon peroxidase showed that the peak intensity was very weak at 402 nm and no new peak appeared, thereby indicating a complete degradation of direct yellow dye.

### 3.7. Physicochemical Characterization of Textile Effluents

The physicochemical parameters of crude textile wastewater before and after fungal treatment separately or in the consortium were investigated. The results showed that the effluent before any treatment had a yellowish color with a conductivity of 1047 µs/cm, temperature of 36 °C (measured by a laboratory thermometer), pH of 8.7, along with BOD, COD, TDS, and TSS values as 342, 611, 1153, and 708 mg·L^−1^, respectively, which indicate a broad range of chemical contaminants (Table 2). The textile effluent properties were highly decreased with the removal percentages of 78.2, 78.4, 58.2, 78.1, and 77.6% for TDS, TSS, conductivity, BOD, and COD, respectively, due to fungal consortium treatment.

The dark yellowish color of crude textile effluent is due to the mixture of chemicals and various dyes used during the dyeing processes [62]. A high pH value (8.7) in the crude textile effluent is mainly due to the use of bicarbonate, carbonate, NaOH, and H_2_O_2_ during the bleaching process in the textile industry [63]. After fungal treatment, the pH value of treated textile effluent decreased in the range of 6.8–7.2. An elevated crude textile wastewater temperature leads to the decreased solubility of gases in water, which is eventually expressed as high BOD/COD values. In this study, the high COD (611 mg·L^−1^) and BOD (342 mg·L^−1^) levels in the textile effluent indicate the high toxicity of the effluents, which is very harmful to the whole ecology and aquatic system of the obtaining water bodies. A high TDS value reduces the sunlight diffusion into the water and eventually reduces the photosynthetic process in the aquatic flora. This process further causes a reduction in the dissolved oxygen value of water bodies, resulting in extremely depleted purification of wastewater by microbes [46]. The treatment of crude textile effluent by the fungal consortium was 78.1 ± 2.1%, which was more than those caused by individual fungal strains after 7 days. These results were similar to those obtained by Saroj et al. [64] who showed that the fungal consortium of three strains, namely *Penicillium oxalicum, Aspergillus niger*, and *Aspergillus flavus*, exhibit a remarkably high potential to degrade the azo dyes (acid red 183, direct blue 15, and direct red 75) at various initial concentrations. Therefore, the growth compatibility of the three strains suggests that none of these were competing against each other in the consortium. Moreover, the reduction in COB, BOD, and color removal of Drimarene blue K_2_RL dyes (50 mg L^−1^) due to treatment by *Aspergillus niger* were 84.7, 85.6, and 71.3% after 24 h [65]. In addition, the efficacy of nine fungal strains, *Fusarium oxysporum* and *Aspergillus niger* being two of them, to decolorizations of three industrial azo-dyes were investigated [66]. Therefore, in this study, a fungal consortium consisting of two fungal strains is suitable for a higher degradation of dyes. The reduction in BOD and COD levels after treatment with the fungal isolates could result from the removal of organic load from the effluent, hence, reducing the toxicity [67]. Reduction in the level of TDS by both isolates and their consortium was in correlation with the works of Ali et al. [68] and Kumar et al. [69] who obtained a similar result.

### 3.8. GC–MS of Untreated and Treated Textile Effluent by a Consortium of Two Fungal Strains

The biodegradable products resulting from the treatment of crude textile effluents by fungal consortium (as the best treatment) were detected using GC–MS as compared with the control (wastewater effluent without any treatments). The intermediate compounds obtained during the biodegradation process were analyzed according to the NIST library [36,37]. For crude textile wastewater before treatment, the GC-MS spectra revealed six major compounds at retention time of 12.5 (1,2,3,4,5-pentamethylcyclopentane), 12.6 (1,2,3,4,5 pentamethylcyclopentene), 14.031 (3,3,4-trimethyldecane), 14.123 (1-methyl-2propylecyclohexane), 14.4 (2,3,4-trimethylpentane), and 31.1 (N-methylbenzeneethanamine) with peaks area 27.8, 7.5, 7.5, 15.9, 9.9, and 13.1%, respectively, as shown in Table 3 and Figure 5A. Data obtained showed that the mass spectra of the treated textile effluent by fungal consortium are diminished to 18.1% (1,2,3,4,5-pentamethylcyclopentane) at retention time 12.5, 4.9% (1,2,3,4,5-pentamethylcyclopentene) at retention time 12.6, 5.5% (3,3,4-timethyldecane) at retention time 14.03, 10.9% (1-methyl,2-propylcyclohexane) at retention time 14.1, and 1.6% (3-ethylpentane) at retention time 14.4, while (N-methyl Benzeneethanamine) was not detected after treatment. Moreover, some peaks appeared because of fungal consortium treatment such as 3.3% (2,4-dimethylheptane) at retention time 7.7, 1.3% (2-Pentenal) at retention time 13.1, 1.6% (3-ethylPentane) at retention time 14.3, 1.8% (3,6-dimethyldecane) at retention time 20.1, and 0.58% (3,7-dimethyldecane) at retention time 22.4 (Table 3, Figure 5B).

The results in the current study were similar to those of Surti and Ansari [54] who found that the GC–MS analysis of metabolites extracted from the MSM broth altered with 0.5-mL dye wastewater showed complete degradation of dye, as indicated by the comparison between GC–MS chromatogram of pure and degraded dyes. Jamil et al. [70] also found that the GC–MS analysis indicates the complete breakdown of disperse violet 26 into inorganic compounds and ions of low molecular weight, for example, H_2_O, CO_2_, NO_3_, etc. Moreover, Fouda et al. [63] reported that biodegradable compounds found in the textile wastewater treated with 1.0 mg of γ-Fe_2_O_3_-NPs∙ mL^−1^ for 6.0 h were analyzed using GC–MS, and compared with those found in the untreated sample. Moreover, the obtained results were similar to those of Salem et al. [15], who reported that the GC–MS spectra of textile effluents before and after treatment were recorded and the potentiality of *Aspergillus niger* was confirmed in textile wastewater treatment.

### 3.9. Phytotoxicity Study

The biodegradation process is promising for the degradation of dyes as well as in the improvement of water quality. However, toxicity is still a mystery [71,72] because biodegradable products may exert more toxicity than the parent dyes molecule; hence, there is always going to be a continual need for toxicity evaluation of treated dyes using reliable and standard analytical method/bioassays. After degradation, the wastewater is disposed of in water bodies that could be used for agricultural purposes. Therefore, it is crucial to assess the phytotoxicity of textile effluents before and after degradation. The relative sensitivities toward the textile effluent and their degradation products concerning *Vicia faba* L. seeds are recorded in Table 4 and Figure 6. Data analysis revealed that the crude textile wastewater reduces the shoot and root length of *V. faba* to 1.2 ± 0.01 and 2.1 ± 0.02 cm, respectively, as compared with the control with shoot length of 17.8 ± 0.7 cm and root length of 7.5 ± 2.1 cm. This reduction can be neglected because the textile wastewater treated with a fungal consortium recorded 15.1 ± 1.01 cm and 6.3 ± 2.1 cm for the shoot and root lengths, respectively.

This study results were similar to those obtained by Ilyas and Rehman [44] who reported that the biodegradable metabolites resulting from the dye degradation by *Aspergillus niger* were not toxic to beneficial microflora and plant growth. Dawkar et al. [73] reported that metabolites produced after decolorization were nontoxic with respect to *Sorghum bicolor* and *Triticum aestivum*. Similarly, Kalyani et al. [74] reported that *Sorghum vulgare* and *Phaseolus mungo* showed a good germination rate as well as a significant growth after irrigating with metabolites extracted from Red BLI dye degradation, as compared to the dye sample. The toxicity of Remazol Black B dye was reduced after treatment with *Zinnia angustifolia* and *Exiguobacterium aestuarii*, as a plant and bacterial remediation, respectively [75]. Biotoxicity and phytotoxicity assays are two major assays that evaluate the toxicity of degraded compounds. Recently, the toxicity evaluation of degraded azo dye direct yellow 4 was reported by using the phytotoxicity assay. Here the phytotoxicity assay showed a considerable decrease in the toxicity of degraded dyes as compared with pure dye [61]. The phytotoxicity of biodegraded textile effluent has also been evaluated before in research [76], with fungal ligninolytic enzymes noted as responsible for its biodegradation potential [77].

## 4. Conclusions

Out of twenty-one fungal strains isolated in this study, two fungal strains identified as *A. flavus* A2 and *F. oxysporium* G2-1 were found to have high potentiality for textile wastewater degradation. This study investigated the decolorization processes of these strains under all optimized conditions of pH, incubation periods, inoculum size, and best external supplementary carbon and nitrogen sources. The optimum conditions for effluent decolorization efficiency of *A. flavus* and *F. oxysporium* were achieved in vitro with glucose, yeast extract supplementation, and 3 discs (7 mm) of fungal inoculum size, at pH 8.0 and 6.0 respectively, and incubation time of 7 days. Significant reduction in color removal, BOD, COD, TSS, and TDS was detected for textile wastewater after treatment by the two fungal strains separately or in a consortium and confirms the successful degradation process. GC–MS and UV-vis spectroscopy confirmed the biodegradation. Phytotoxicity of textile effluents before and after degradation upon *Vicia faba* L seed germinations also confirmed the biodegradation process. Finally, this study concluded that bio-treatment of textile wastewater by fungal cell provided a cost-effective, easily applicable, and eco-friendly method.

## Figures and Tables

**Figure 1 jof-07-00193-f001:**
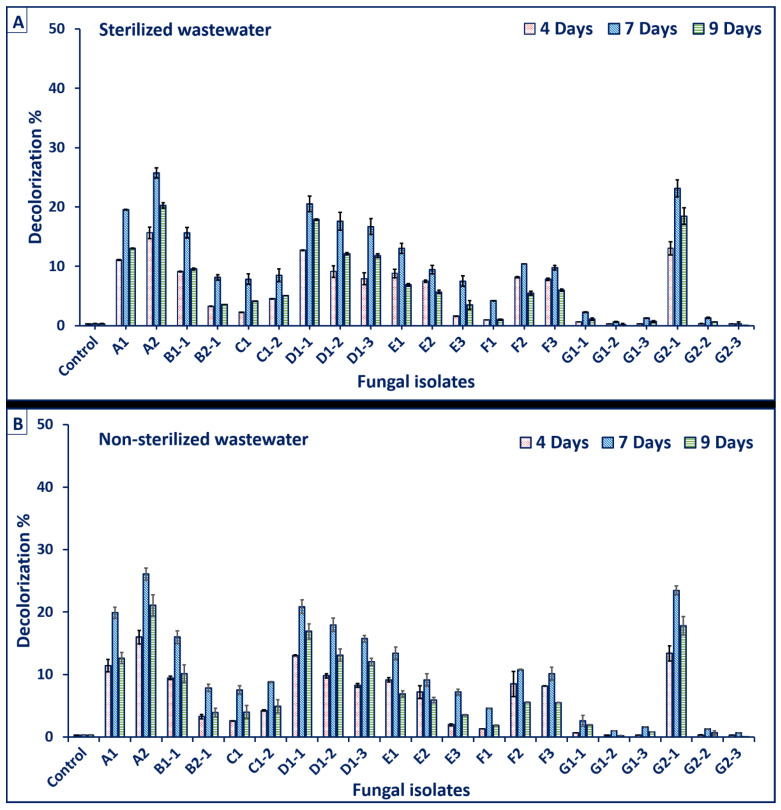
Assessment of the efficacy of isolated fungal strains to decolorization of textile wastewater under sterilized condition (**A**) and non-sterilized condition (**B**). See results for the identities of the fungal isolates tested in this study.

**Figure 2 jof-07-00193-f002:**
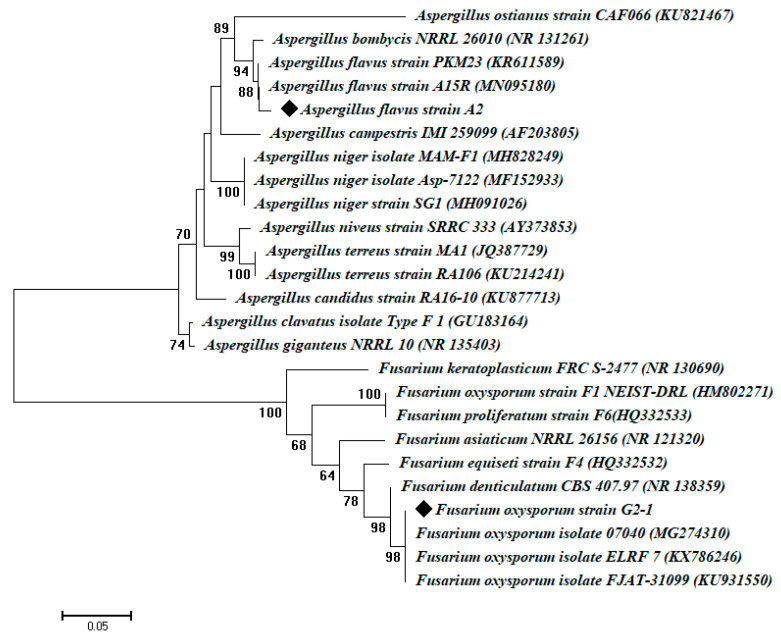
Phylogenetic tree of ITS sequences of the fungal isolates A2 and G2-1 with the sequences from NCBI and designated as *Aspergillus flavus* A2 and *Fusarium oxysporum* G2-1.

**Figure 3 jof-07-00193-f003:**
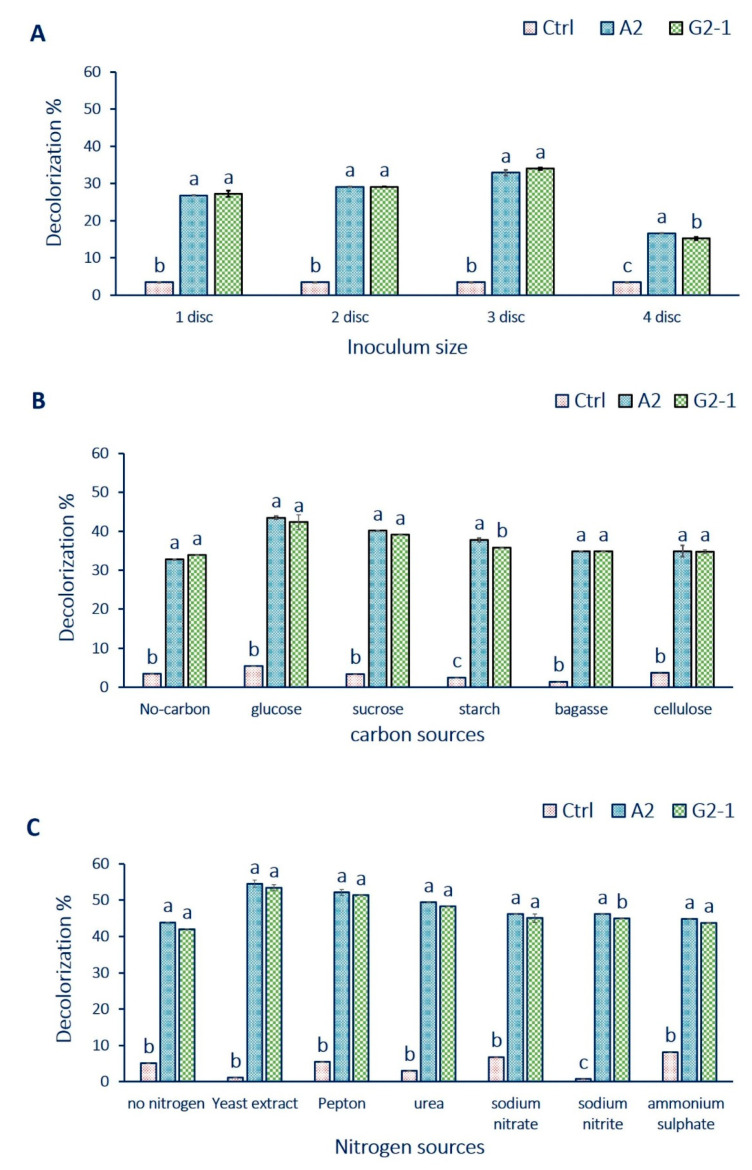
The optimization process of decolorization of crude textile wastewater using fungal strains. (**A**) denotes the effect of different inoculum sizes; (**B**) denotes the effect of different carbon sources; (**C**) the effect of different nitrogen sources. No carbon or no nitrogen denotes inoculated media without adding external carbon or nitrogen source Error bars are Mean ± SE (*n* = 3). Ctrl, control without fungal inoculation; *A. flavus* (A2) and *F. oxysporum* (G2-1). Different letters on bars denote that mean values are significantly different (*p* ≤ 0.05) by the Tukey test.

**Figure 4 jof-07-00193-f004:**
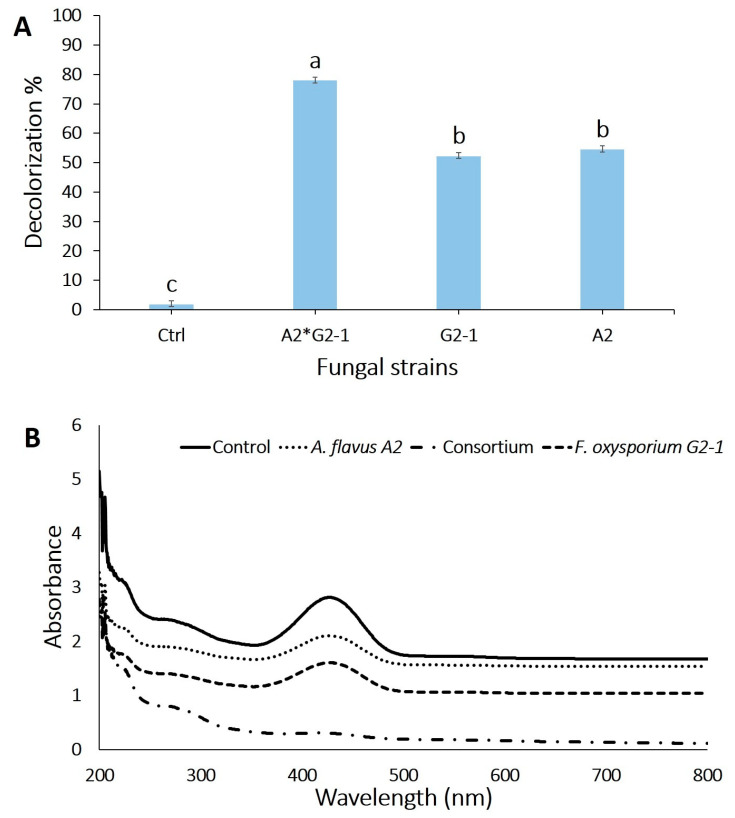
Assessment decolorization efficacy of crude textile wastewater using individual fungal strain and their consortium. (**A**) Decolorization percentages, Ctrl, without fungal inoculation; A2, *Aspergillus flavus;* G2-1, *Fusarium oxysporium*; A2*G2-1, a mixture of *A. flavus* with *F. oxysporium*. (**B**) UV-Visible spectrum for textile effluent before and after optimization and treated by either individual or consortium fungal strains. Error bars are Mean ± SE (*n* = 3). Different letters on bars denote that mean values are significantly different (*p* ≤ 0.05) by the Tukey test.

**Figure 5 jof-07-00193-f005:**
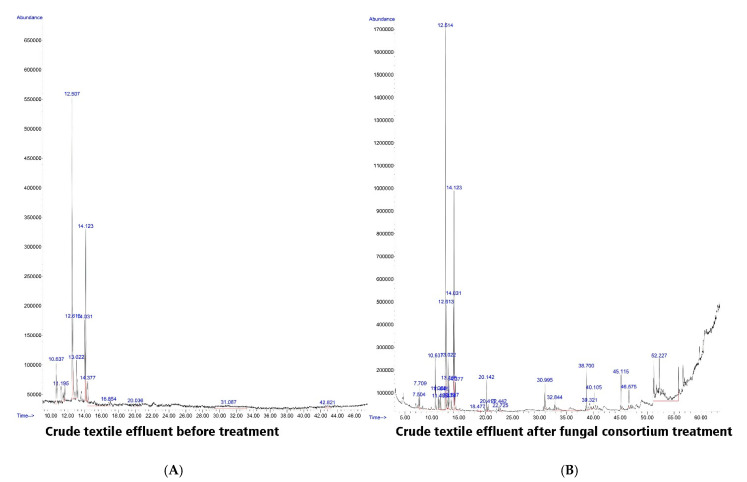
GC-mass spectra of textile wastewater before (**A**) and after (**B**) fungal consortium treatment.

**Figure 6 jof-07-00193-f006:**
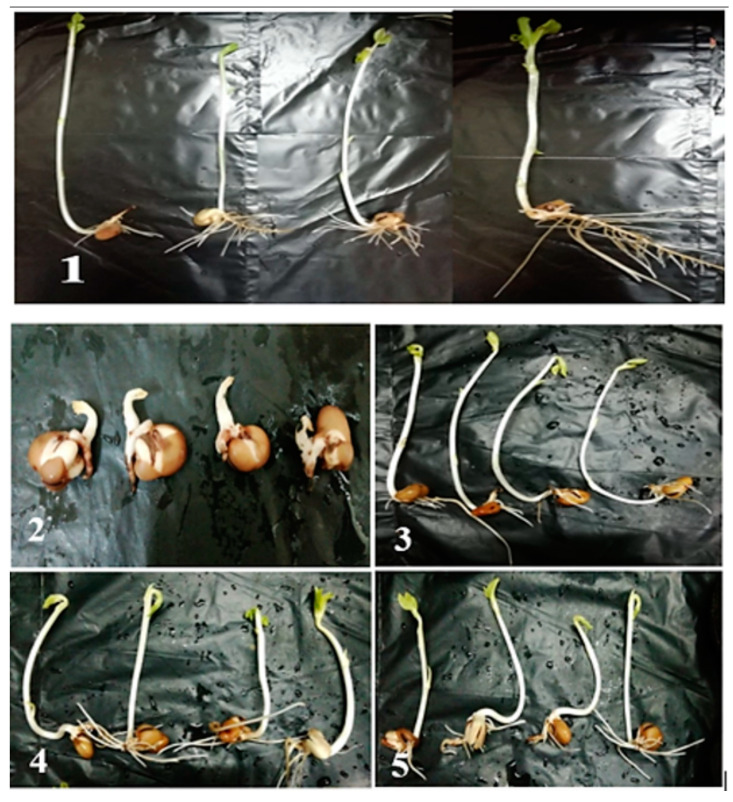
(**1**) Phytotoxicity study of textile wastewater on *Vicia faba* L. seeds using two fungal strains and their consortium. (**1**) Seeds after being treated with freshwater, (**2**) Seeds after being treated with textile effluent without fungal treatment: (**3**) Seeds after being treated with textile effluent degraded by a fungal consortium (*A. flavus* with *F. oxysporium* A2*G2-1). (**4**) Seeds after being treated with textile effluent degraded by *A. flavus*; A2 (**5**) Seeds after being treated with textile effluent degraded by *F. oxysporium* G2-1.

**Table 1 jof-07-00193-t001:** Effect of different pH values on decolorization of textile wastewater by *A. flavus* (A2), *F. oxysporum* (G2-1), and without fungal inoculation (Ctrl) after different incubation periods, means ± SE (*n* = 3).

pH	Biodegradation (%) of Textile Wastewater without Fungal Inoculation (Ctrl)						
	Incubation Periods (Days)						
	3	4	5	6	7	8	9
6	0.4 ± 0.0	0.2 ± 0.01	0.4 ± 0.0	0.2 ± 0.01	0.3 ± 0.0	0.4 ± 0.0	0.4 ± 0.01
7	0.3 ± 0.0	0.8 ± 0.0	0.3 ± 0.01	0.3 ± 0.0	0.3 ± 0.0	0.3 ± 0.0	0.3 ± 0.0
8	0.4 ± 0.01	1.2 ± 0.02	0.4 ± 0.0	0.3 ± 0.0	0.4 ± 0.01	0.4 ± 0.0	0.4 ± 0.0
9	0.5 ± 0.0	0.8 ± 0.1	0.4 ± 0.0	0.3 ± 0.0	0.3 ± 0.0	0.3 ± 0.01	0.5 ± 0.01
10	0.9 ± 0.01	0.3 ± 0.0	0.9 ± 0.1	0.3 ± 0.0	0.4 ± 0.0	0.3 ± 0.01	0.9 ± 0.0
11	0.7 ± 0.01	0.7 ± 0.0	0.9 ± 0.0	0.4 ± 0.0	0.3 ± 0.01	0.3 ± 0.0	0.8 ± 0.01
**pH**	**Biodegradation (%) of Textile Wastewater by *A. flavus* (A2)**						
	**Incubation Periods (Days)**						
	**3**	**4**	**5**	**6**	**7**	**8**	**9**
6	1.9 ± 0.0	3.6 ± 0.1	5.4 ± 0.3	6.3 ± 0.6	7.4 ± 0.4	4.2 ± 0.3	0.2 ± 0.0
7	3.1 ± 0.0	6.6 ± 0.02	8.4 ± 0.1	11.9 ± 0.4	16.5 ± 0.4	8.6 ± 0.1	2.2 ± 0.01
8	7.3 ± 0.0	11.9 ± 0.01	16.5 ± 0.1	19.8 ± 0.2	26.2 ± 1.3	17.5 ± 1.7	4.2 ± 0.1
9	3.4 ± 1.2	4.5 ± 1.02	7.4 ± 0.3	10.2 ± 1.02	11.3 ± 0.3	8.03 ± 0.2	1.1 ± 0.01
10	1.9 ± 1.01	2.3 ± 1.0	5.4 ± 0.7	6.3 ± 1.2	7.4 ± 1.02	4.2 ± 0.4	0.2 ± 0.01
11	0.4 ± 0.2	2.3 ± 0.2	3.4 ± 0.9	5.2 ± 0.3	6.2 ± 1.4	2.3 ± 0.01	0.1 ± 0.2
**pH**	**Biodegradation (%) of Textile Wastewater by *F. oxysporum* (G2-1)**						
	**Incubation Periods (Days)**						
	**3**	**4**	**5**	**6**	**7**	**8**	**9**
6	7.4 ± 0.0	13.8 ± 0.4	18.6 ± 0.4	21.5 ± 0.4	27.8 ± 0.02	13.8 ± 0.02	2.6 ± 0.01
7	6.8 ± 0.02	7.4 ± 0.1	16.4 ± 0.1	19.9 ± 0.7	25.7 ± 0.02	13.8 ± 0.04	6.6 ± 0.03
8	5.9 ± 0.1	7.4 ± 0.02	13.6 ± 0.4	18.2 ± 0.3	23.7 ± 0.4	10.6 ± 0.1	2.2 ± 0.01
9	3.04 ± 0.1	3.7 ± 0.1	6.1 ± 0.3	7.3 ± 0.4	13.8 ± 0.01	4.2 ± 0.03	0.2 ± 0.0
10	0.2 ± 0.0	6.5 ± 0.1	1.1 ± 0.1	3.04 ± 0.3	2.2 ± 0.2	0.2 ± 0.02	0.2 ± 0.0
11	0.7 ± 0.0	0.22 ± 0.1	0.4 ± 0.0	0.3 ± 0.0	0.2 ± 0.0	0.1 ± 0.0	0.1 ± 0.0

**Table 2 jof-07-00193-t002:** Physicochemical characteristics of a textile effluent before and after treatment using the individual and consortium fungal strains.

Parameters	Textile Wastewater Treated by							
	Ctrl.	Removal Percentages (%)	*A. flavus* A2	Removal Percentages (%)	*F. oxysporium* G2-1	Removal Percentages (%)	Consortium (A2 + G2-1)	Removal Percentages (%)
pH	8.7 ± 0.1	-	7.3 ± 0.1	-	7.2 ± 0.4	-	6.8 ± 0.7	-
TDS (mg·L^−1^)	1153 ± 1.2	-	528 ± 1.4	54.2	540 ± 0.9	53.2	251 ± 0.4	78.2
TSS (mg·L^−1^)	708 ± 0.9	-	321 ± 0.7	54.7	335 ± 0.01	52.7	153 ± 0.3	78.4
Conductivity (µs·cm^−1^)	1047 ± 1.3	-	623 ± 1.6	40.5	680 ± 0.1	35.1	438 ± 0.01	58.2
BOD (mg·L^−1^)	342 ± 0.9	-	153 ± 0.7	55.3	161 ± 0.4	52.9	75 ± 0.8	78.1
COD (mg·L^−1^)	611 ± 0.3	-	274 ± 0.3	55.2	285 ± 0.5	53.4	137 ± 0.3	77.6

Ctrl, textile wastewater effluent without fungal inoculation; Error bars are Mean ± SE (*n* = 3).

**Table 3 jof-07-00193-t003:** Suggested compounds with peak and mass area resulted from GC –MS of textile effluent before and after fungal consortium treatment.

Retention Time (min.)	Suggested Name	Control		Treated by a Fungal Consortium	
		Peak area %	Mass area %	Peak Area %	Mass Area %
12.5	1,2,3,4,5-pentamethylcyclopentane	27.8	100.0	18.1	64.3
12.6	1,2,3,4,5 pentamethyl cyclopentene	7.5	27.01	4.9	17.7
14.03	3,3,4-trimethyldecane	7.5	27.1	5.5	19.3
14.1	1-methyl,2-propylcyclohexane	15.9	56.9	10.9	38.8
14.4	2,3,4-trimehylpentane	9.9	35.5	1.6	5.7
31.1	N-methyl Benzeneethanamine	13.1	47.2	-	-
7.7	2,4-dimethylheptane	-	-	3.3	11.6
13.2	2-Pentenal	-	-	1.3	4.7
14.4	3-ethylPentane	-	-	1.6	5.7
20.1	3,6-dimethyldecane	-	-	1.8	6.2
22.4	3,7-dimethyldecane	-	-	0.6	2.0

-, denotes not detected.

**Table 4 jof-07-00193-t004:** Phytotoxicity study of different textile wastewater on soybean seeds using the two fungal strains and their consortium.

Parameters	Length of Shoot and Root (cm) after 7 Days	
	Shoot Length (cm)	Root Length (cm)
Water	17.8 ± 0.7 ^a^	7.5 ± 2.1 ^a^
Ctrl	1.2 ± 0.01 ^d^	2.1 ± 0.02 ^c^
A2	7.3 ± 0.6 ^c^	2.96 ± 0.3 ^bc^
G2-1	6.1 ± 0.2 ^c^	5.3 ± 0.7 ^b^
A2*G2-1	15.1 ± 1.01 ^b^	6.3 ± 2.1 ^a^

Ctrl, textile effluent without fungal inoculation; A2, *Aspergillus flavus;* G2-1, *Fusarium oxysporium;* A2*G2-1, the mixture of *Aspergillus flavus* with *Fusarium oxysporium*. Different letters between columns denote that mean values are significantly different (*p* ≤ 0.05) by Tukey test, means ± SE (*n* = 4).

## Data Availability

The data presented in this study are available on request from the corresponding author.

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
