# Peer review of "Biological Treatment of Real Textile Effluent Using Aspergillus flavus and Fusarium oxysporium and Their Consortium along with the Evaluation of Their Phytotoxicity"

_jof, 2021, doi:10.3390/jof7030193_

Round 1
Reviewer 1 Report
biological treatment or decolorization of effluents is a very important aspect in the textile industry
The effluents released from the textile industry are considered as one of the highest liquid pollutants, based on the published literature. About 280,000 tons of textile dyes are discharged as wastes in textile effluents every year worldwide. A massive quantity of water is consumed during the textile processing steps such as washing, dyeing, seizing, and others. Approximately, 10–20 L of pure water is consumed for dyeing 1 kg of fabric.
Why authors did not mention that in the future dyes from chemistry should be totally forbidden? Sustainability is the KEYWORD. Before the chemistry era, humans were using natural dyes for textile. Time is now to go back to Nature.
(natural pigments are easily broken down)
literature searches
textile x effluent 7425 refs
biological treatment x textile effluent 1086
2 with Aspergillus flavus, 2 with Fusarium oxysporum
Do you have any explanation about this? such fungal strains not investigated before? fungi less efficient than bacteria?
To my opinion, decolorization is not enough, the target is not to produce uncolored effluents, but fully degraded effluents (an uncolored textile effluent can still be toxic for Nature)
GC–MS Spectroscopy for untreated/treated wastewater
I would recommend the use of LC-HRMS/MS
Water ResearchVolume 190, 15 February 2021, Article number 116745
Characterization of water-soluble synthetic polymeric substances in wastewater using LC-HRMS/MS(Article)(Open Access)
- Mairinger, T.aEmail Author,
- Loos, M.b,
- Hollender, J.a,cEmail Author
- View Correspondence (jump link)
Synthetic water-soluble polymeric materials are widely employed in e.g. cleaning detergents, personal care products, paints or textiles. Accordingly, these compounds reach sewage treatment plants and may enter receiving waters and the aquatic environment. Characteristically, these molecules show a polydisperse molecular weight distribution, comprising multiple repeating units, i.e. a homologous series (HS). Their analysis in environmentally relevant samples has received some attention over the last two decades, however, the majority of previous studies focused on surfactants and a molecular weight range <1000 Da. To capture a wider range on the mass versus polarity plane and extend towards less polar contaminants, a workflow was established using three different ionization strategies, namely conventional electrospray ionization, atmospheric pressure photoionization and atmospheric pressure chemical ionization. The data evaluation consisted of suspect screening of ca. 1200 suspect entries and a non-target screening of HS with pre-defined accurate mass differences using ca. 400 molecular formulas of repeating units of HS as input and repeating retention time shifts as HS indicator. To study the fate of these water-soluble polymeric substances in the wastewater treatment process, the different stages, i.e. after primary and secondary clarifier, and after ozonation followed by sand filtration, were sampled at a Swiss wastewater treatment plant. Remaining with two different ionization interfaces, ESI and APPI, in both polarities, a non-targeted screening approach led to a total number of 146 HS (each with a minimum number of 4 members), with a molecular mass of up to 1200 detected in the final effluent. Of the 146 HS, ca 15% could be associated with suspect hits and approximately 25% with transformation products of suspects. Tentative characterization or probable chemical structure could be assigned to almost half of the findings. In positive ionization mode various sugar derivatives with differing side chains, for negative mode structures with sulfonic acids, could be characterized. The number of detected HS decreased significantly over the three treatment stages. For HS detectable also in the biological and oxidative treatment stages, a change in HS distribution towards to lower mass range was often observed.
for Aspergillus flavus, ref Saroj (59) is properly cited
missing ref could be
Environmental Science and Pollution ResearchVolume 19, Issue 5, June 2012, Pages 1728-1737
An investigation of anthraquinone dye biodegradation by immobilized Aspergillus flavus in fluidized bed bioreactor(Article)
- Andleeb, S.a,bEmail Author,
- Atiq, N.a,
- Robson, G.D.c,
- Ahmed, S.a
missing ref for Fusarium oxysporum
Industria TextilaVolume 67, Issue 3, 2016, Pages 181-188
Evaluation of decolonisation abilities of natural fungal isolates(Article)
- Iordache, O.a,bEmail Author,
- Cornea, C.P.a,
- Popa, G.a,
- Dumitrescu, I.b,
- Diguja, C.a,
- Varzaru, E.b,
- Rodino, S.a,c,
- Ionescu, I.a,d,
- Matei, A.a,e
View additional affiliations
Abstract View references (33)
Nine previously isolated and identified fungal strains from post-finishing textile effluents were investigated for their decolorisation ability of three Bemacid textile dyes, in aqueous solution. Identified fungal isolates belong to the following groups: Trichoderma parceramosum, Trichoderma reesei, Trichoderma longi, Polyporus squamosus and Fusarium oxysporum, along with Aspergillus Niger (IMl 45551), used as a reference collection strain. Maximum absorbance peaks in visible region, for each dye, were assessed spectrophotometrically and dye residual concentration reduction were assessed at 500nm for Bemacid ROT, 370nm Bemacid GELB and 590nm Bemacid BLAU. Purity screening of the dyes was assessed by Thin Layer Chromatography (TLC), and decolorisation assays were carried out in nutritive media, in 5 simultaneous batches, for 3, 5, 9, 12 and 15 days, with each batch run in triplicate, and results expressed as mean of triplicate values for each combination of strain and dye. Quantitative analysis of solutions decolorisation was carried out via UV-VIS spectrophotometry assessment, quantifying decolorisation degree over post-incubation period at 29°C, each dye residual concentration reduction ranging from 20.98% to 98.02% for Bemacid ROT, 43.5% to 96.06% for Bemacid GELB and 35.68% to 98.38% for Bemacid BLAU, thus promoting biological approach of wastewater treatment with the aid of filamentous fungi as an efficient, cost effective and environmental friendly solution.
Author Response
Thank you very much for reviewing our manuscript, your agreement, and your valuable comments.
Response to Reviewers Report-1
(jof-1140105)
To Reviewer #1
Thank you very much for reviewing our manuscript, your agreement, and valuable comments. We are also grateful for the favorable comments. The manuscript was undergoing English editing in enago English editing service (www.enago.com). We made corrections and we hope they meet with your approval. The detailed explanation is given below.
Reviewer comment #: biological treatment or decolorization of effluents is a very important aspect in the textile industry.
The effluents released from the textile industry are considered as one of the highest liquid pollutants, based on the published literature. About 280,000 tons of textile dyes are discharged as wastes in textile effluents every year worldwide. A massive quantity of water is consumed during the textile processing steps such as washing, dyeing, seizing, and others. Approximately, 10–20 L of pure water is consumed for dyeing 1 kg of fabric.
Why authors did not mention that in the future dyes from chemistry should be totally forbidden? Sustainability is the KEYWORD. Before the chemistry era, humans were using natural dyes for textile. Time is now to go back to Nature.
(natural pigments are easily broken down)
Author response #: Thank you very much for your comment. We clarified these meaning in abstract as follows “Finally, we recommended the decrease of excessive uses of synthetic dyes and utilized biological approaches for treatment of real textile effluents to reuse in irrigation of uneaten plants especially with water scarcity worldwide.”.
Reviewer comment #: literature searches
textile x effluent, 7425 refs
biological treatment x textile effluent, 1086
2 with Aspergillus flavus, 2 with Fusarium oxysporum
Do you have any explanation about this? such fungal strains not investigated before? fungi less efficient than bacteria?
Author response #: Thank you for your comment. Really, low number studies were published about using fungi in biological treatment, but this not meaning its less efficient than bacteria. Fungi are eukaryotic and have large surface area exposed to dyes, thus the adsorption process using fungi is better than bacteria. Moreover, fungi are characterized by their huge number of enzymes and organic compounds can be used for biodegradation of pollutants. Also, fungi are characterized by their ability to tolerate environmental stress higher than bacteria.
Reviewer comment #: To my opinion, decolorization is not enough, the target is not to produce uncolored effluents, but fully degraded effluents (an uncolored textile effluent can still be toxic for Nature)
Author response #: Thank you for your comment. the efficacy of fungal species to degrade textile effluent were assessed by GC-MS and the toxicity of treated effluents compared with untreated was investigated using phytotoxicity on Vicia faba. Data showed that the shoot length after irrigation with effluents treated by fungal consortium was 15.12 ± 1.01cm as compared with that treated by tap-water, which was 17.8 ± 0.7cm. data indicates that, although color not completely disappeared, but the toxicity was decreased as compared with untreated.
Reviewer comment #: GC–MS Spectroscopy for untreated/treated wastewater
I would recommend the use of LC-HRMS/MS
Author response #: Thank you for your comment. we will take this analysis in the future study, but in the current time, we cannot repeat this test using LC-HRMS/MS due to spread of Covid-19 pandemic and the lab is closed.
Reviewer comment #: for Aspergillus flavus, ref Saroj (59) is properly cited.
missing ref could be.
Environmental Science and Pollution ResearchVolume 19, Issue 5, June 2012, Pages 1728-1737. An investigation of anthraquinone dye biodegradation by immobilized Aspergillus flavus in fluidized bed bioreactor.
missing ref for Fusarium oxysporum
Industria TextilaVolume 67, Issue 3, 2016, Pages 181-188. Evaluation of decolonisation abilities of natural fungal isolates
Author response #: Thank you for your comment. the recommended references were cited as follows “Moreover, the reduction in COB, BOD, and color removal of Drimarene blue K2RL dyes (50 mg L-1) due to treatment by Aspergillus niger were 84.7, 85.6, and 71.3% after 24h [65]. Also, the efficacy of nine fungal strains, Fusarium oxysporum and Aspergillus niger two of them, to decolorizations of three industrial azo-dyes were investigated [66].”
Finally, we hope the response meets the reviewer approve.
Reviewer 2 Report
The manuscript entitled “Biological treatment of real textile effluent using Aspergillus flavus and Fusarium oxysporium and their consortium along with the evaluation of their phytotoxicity is well written and presented interesting findings. The manuscript can be accepted with the following changes.
Comments:
- In the abstract authors should add recommendations for this work.
- Similar worlds with title should be deleted as of a keyword.
- Line 191 “mL.min−1” pls check
- Line 226 authors used space between values and SD whereas line 251 without space, pls check throughout MS and format similar and in tables.
- Recommended to avoid to use “our data revealed (L 229) “Our results showed (L 249), Our results also showed (L 253), Our finding supports (L 262), Our data showed (L465), Our results (L 475, 482) so on.
- Line 295 compatible with that of [15]?
- Table 1 most of the cases SE is o or 0.01?
- Table one last row column 3 what author indicate by superscript zero?
- In the table, decimal digits should be uniform.
- Use abbreviated form or fungi.
- Figure 5 is not visual.
Author Response
Thank you very much for reviewing our manuscript, your agreement, and valuable comments.
Response to Reviewers Report-2
(jof-1140105)
Thank you very much for reviewing our manuscript. We are also grateful for your valuable positive comments. The manuscript was undergone to English editing at enago for English editing (www.enage.com). A detailed explanation is below.
Reviewer comment #: The manuscript entitled “Biological treatment of real textile effluent using Aspergillus flavus and Fusarium oxysporium and their consortium along with the evaluation of their phytotoxicity is well written and presented interesting findings. The manuscript can be accepted with the following changes.
Author response #: thank you for your agreement and comment.
Reviewer comment #: In the abstract authors should add recommendations for this work.
Author response #: thank you for your comment. As reviwer commended we adding final recommendation in abstract as follows “ Finally, we recommended the decrease of excessive uses of synthetic dyes and utilized biological approaches for treatment of real textile effluents to reuse in irrigation of uneaten plants especially with water scarcity worldwide.”
Reviewer comment #: Similar worlds with title should be deleted as of a keyword.
Author response #: thank you for your comment. the similar words between title and keyword were revised.
Reviewer comment #: Line 191 “mL.min−1” pls check
Author response #: we thanks reiviwer for your observation. It is correct.
Reviewer comment #: Line 226 authors used space between values and SD whereas line 251 without space, pls check throughout MS and format similar and in tables.
Author response #: we thanks reiviwer for your observation. It is correct and standardized in throughout manuscript.
Reviewer comment #: Recommended to avoid to use “our data revealed (L 229) “Our results showed (L 249), Our results also showed (L 253), Our finding supports (L 262), Our data showed (L465), Our results (L 475, 482) so on.
Author response #: we thanks reiviwer for your observation. It is correct throughout the manuscript.
Reviewer comment #: Line 295 compatible with that of [15]?
Author response #: thank you for your comment. it is correct as follows “compatible with that of Salem et al. [15].
Reviewer comment #: Table 1 most of the cases SE is o or 0.01?
Author response #: thank you for your comment. this is because of our data were unnormal distributed and we did ANOVA analysis to these original data.
Reviewer comment #: Table one last row column 3 what author indicate by superscript zero?
Author response #: please accept my applogy, it is a wrongly placed number and we delete it.
Reviewer comment #: In the table, decimal digits should be uniform.
Author response #: thank you for your comment. it is correct throughout the manuscript.
Reviewer comment #: Use abbreviated form or fungi
Author response #: thank you for your comment. it is correct throughout the manuscript.
Reviewer comment #: Figure 5 is not visual.
Author response #: thank you for your comment. the figure was improved and increased their resolution as shown in manuscript
Finally, we hope the response meets the reviewer approve.